



# Past fire dynamics inferred from polycyclic aromatic hydrocarbons and monosaccharide anhydrides in a stalagmite from the archaeological site of Mayapan, Mexico

Julia Homann[1], Niklas Karbach[1], Stacy A. Carolin[2,3], Daniel H. James[2], David Hodell[2], Sebastian F. M.
Breitenbach[4], Ola Kwiecien[4], Mark Brenner[5], Carlos Peraza Lope[6], Thorsten Hoffmann[1]

[1]Department of Chemistry, Johannes Gutenberg-Universität, Mainz, Germany
[2]Department of Earth Sciences, University of Cambridge, Cambridge, UK
[3]School of Archaeology, University of Oxford, Oxford, UK
[4]Department of Geography and Environmental Sciences, Northumbria University, Newcastle upon Tyne, UK
[5]Department of Geological Sciences, University of Florida, FL, USA
[6]Instituto Nacional de Antropología e Historia, Centro INAH Yucatán, Mérida, México

*Correspondence to*: Thorsten Hoffmann (hoffmant@uni-mainz.de)

## Abstract

Speleothems (cave stalagmites) contain inorganic and organic substances that can be used to infer past changes in local and regional paleoenvironmental conditions. Specific biomarkers can be employed to elucidate the history of past fires, caused by interactions among climate, regional hydrology, vegetation, human and fire activity. We conducted a simple solid-liquid extraction on pulverised carbonate samples to prepare them for analysis of 16 polycyclic aromatic hydrocarbons (PAHs) and three monosaccharide anhydrides (MAs). The preparation method requires only small samples (0.5-1.0 g); PAHs and MAs were measured by GC-MS and LC-HILIC-MS, respectively. Detection limits range from 0.05-2.1 ng for PAHs and 0.01-0.1 ng for MAs. We applied the method to 10 samples from a ~400-year-old stalagmite from Cenote Ch'en Mul, at Mayapan, the largest Postclassic Maya capital of the Yucatán Peninsula. We found a strong correlation (r=0.75, p < 0.05) between the major MA (levoglucosan) and non-alkylated PAHs (Σ15). We investigated multiple diagnostic PAH and MA ratios and found that although not all were applicable as paleo-fire proxies, ratios that combine PAHs with MAs are promising tools for identifying different fire regimes and inferring the type of fuel burned. In the 1950s and 1960s, levoglucosan and Σ15 concentrations roughly doubled compared to other times in the last 400 years, suggesting greater fire activity at Mayapan during these two decades. The higher concentrations of fire markers may be due to land clearance at the site and explorations of the cave by the Carnegie Institution archaeologists.

## 1 Introduction

Speleothems are valuable continental paleoenvironmental archives. They can grow continuously over long time periods (Fairchild und Baker 2012; Gałuszka et al. 2017) and can provide very high-resolution (sub-annual) proxy time series of past



climate and environment (Ridley et al. 2015; Braun et al. 2023). Most can be reliably dated using uranium series isotopes (Mason et al. 2022; Scholz und Hoffmann 2008). Well-established inorganic proxies such as $\delta^{18}O$ and $\delta^{13}C$ are increasingly complemented by organic biomarkers that record more specific aspects of the paleoenvironmental conditions (Baker et al.

2019; Blyth et al. 2016; Bosle et al. 2014; Heidke et al. 2019; Homann et al. 2022). One important aspect of past environments is the occurrence and dynamics of fires, whether natural or anthropogenic. The development of fire-sensitive proxies in paleoenvironmental archives can help elucidate interactions among climate, regional hydrology, vegetation, and fire activity (Campbell et al. 2023). Two such fire proxies are polycyclic aromatic hydrocarbons (PAHs) and monosaccharide anhydrides (MAs). PAHs are products of incomplete combustion of biomass and petrogenic fuels over a wide temperature range

(200-700 °C) (Han et al. 2020; Tobiszewski und Namieśnik 2012; Yunker et al. 2002b; McGrath et al. 2003). The presence of specific biomarkers is indicative of the fuel source. Retene, for example, is a PAH that is a unique marker for the combustion of gymnosperm biomass (Ramdahl 1983; Wakeham et al. 1980). MAs are formed only during combustion of biomass at lower temperatures (150-350 °C). The predominant MA is levoglucosan, which is formed during combustion of cellulose (Elias et al. 2001; Simoneit 2002). Both PAHs and MAs can be present in the gas and the particle phase (Lammel et al. 2009; Xie et al.

2014) and have been reported to undergo long-range transport (Luo et al. 2020; Zennaro et al. 2014).Their atmospheric residence times, however, differ widely, ranging from 1-3 hours (gas phase) to 4-5 days (particulate phase) for PAHs and 2-26 days for MAs (Bai et al. 2013; Fraser und Lakshmanan 2000; Slade und Knopf 2013). This difference in atmospheric longevity may explain for why Denis et al. (2012) found PAHs to record only very local (≤0.5 km) fires, whereas known fires that occurred 1-2 km from the sampling site were not recorded.

These different behaviours of PAHs and MAs are the rationale for analysis of both fire proxies in tandem, not only to detect the presence of fire but also to explore changes in fire regime (e.g. fire frequency, intensity, and fuel source). Whereas PAHs and MAs have been measured simultaneously in lake sediment cores (Argiriadis et al. 2018; Battistel et al. 2017; Callegaro et al. 2018), they have not yet been investigated jointly in speleothems, although both have been extracted individually from stalagmites (Argiriadis et al. 2019; Homann et al. 2022; Perrette et al. 2008).

Another motivation to survey different markers simultaneously is to better understand their transport and incorporation mechanisms in speleothem carbonate. In principle, PAHs and MAs can be transported into a stalagmite either via infiltrating water or via deposited aerosol particles via cave ventilation or by cave-internal sources, respectively. Whereas samples from caves without substantial ventilation (i.e. with no or very narrow entrances) would only archive a dripwater-derived signal, it is likely that in caves with significant ventilation and easy human access, markers would be introduced via both dripping water

and externally introduced or cave internally formed aerosol particles.

Here we focus on the extraction of PAHs and MAs from speleothem carbonate. We report the results of sequential extraction of 16 PAHs and three MAs from a speleothem and their analysis using GC-MS and LC-HILIC-MS, respectively. We applied the new, simple method to a young (~400-year-old) stalagmite (MAYA-22-7) collected in August 2022 from Cenote Ch'en Mul, Mayapan, Yucátan Peninsula, Mexico. Several smaller stalagmites from the same cave contained charcoal inclusions,

indicating that the studied stalagmite was a promising candidate for this proof-of-concept study. We used the measured PAH





and MA concentrations and selected diagnostic ratios to interpret past aspects of fire dynamics in the cave and surrounding area.

## 2 Methods

### 2.1 Materials

Ultrapure methanol (MeOH, LC/MS grade) was obtained from Carl Roth, ultrapure dichloromethane (DCM, LC/MS grade, ≥99.8 %) and water (LC/MS grade) were purchased from Fisher Scientific. Ultrapure acetonitrile (ACN, LC/MS grade) and ammonium formiate (99 %) were obtained from VWR Chemicals. Analytical standards of levoglucosan (1,6-Anhydro-$\beta$-D-glucopyranose, 99 %), $p$-terphenyl (≥99 %), naphthalene-$d_8$ (2000 µg mL$^{-1}$ in DCM), acenaphthene-$d_{10}$ (2000 µg mL$^{-1}$ in DCM), phenanthrene-$d_{10}$ (2000 µg mL$^{-1}$ in DCM), and a standard of 16 PAHs (QTM PAH Mix, 2000 µg mL$^{-1}$ in DCM), as

well as ultrapure ethyl acetate (EA, GC/MS grade) were purchased from Sigma-Aldrich. An analytical standard of $^{13}C_6$-levoglucosan (98 %) was obtained from Cambridge Isotope Laboratories. An analytical standard of retene (10 µg mL$^{-1}$ in cyclohexane) was purchased from LGC and analytical standards of mannosan (1,6-Anhydro-$\beta$-D-mannopyranose) and galactosan (1,6-Anhydro-$\beta$-D-galactopyranose) were obtained from Carbosynth Ltd. Ultrapure water with 18.2 MΩ resistance was produced using a Milli-Q water system from Merck Millipore. Solid-phase extraction columns (CHROMABOND SiOH,

3 mL tubes, 45 µm particle size) were purchased from Macherey-Nagel.

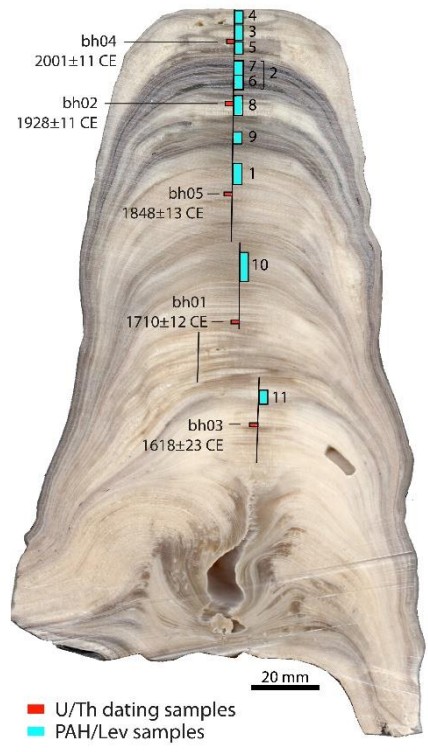



**Figure 1: Scan of stalagmite MAYA-22-7 with samples taken for U-Th dating and biomarker analysis indicated in red and cyan, respectively. For simplicity, rectangles indicate the position of the samples projected to the central growth axis, whereas samples were drilled off-axis, along growth layers. Black lines indicate the growth axis along which grey values were measured. Ages are given in calendar years (C.E. = Common Era) with 95% confidence intervals.**

The developed extraction and measurement methods were tested on stalagmite MAYA-22-7, collected in 2022 from Cenote Ch'en Mul at the Postclassic Maya archaeological site of Mayapan in northern Yucatan (Mexico), with permission granted by the Instituto Nacional de Antropología e Historia (INAH). The stalagmite was retrieved from a narrow passage in the southeast sector of the cave (Fig. S1). The growth surface of stalagmite MAYA-22-7 was wet from drip water prior to collection. The stalagmite was cut in half lengthwise, i.e. along the growth axis, and cut again to produce a central working slab and an off-axis working piece, using a Diamond WireTec DSW175 wire saw with a 0.35-mm-diameter diamond-studded steel wire at Northumbria University, UK. Powdered samples for biomarker analysis and U-Th dating were drilled at the University of Cambridge along exposed growth layers on the working half (Fig. 1), using a handheld Dremel drill with a 0.8-mm-diameter tungsten carbide dental drill bit. The bench surface, drill, drill bit, and aluminium tools for powder collection were all sprayed with compressed air and wiped down with a Kimwipe™ and methanol between sample drilling. Sample pit vertical heights along the growth axis differed depending on the sample pit size, with a range of 3 to 8 mm (Fig. 1). The stalagmite surface was also wiped with a Kimwipe™ and methanol before each pit was drilled. Powder sample weights ranged from 500 to 1200 mg and were stored in methanol-cleaned glass sample vials with a plastic screw cap.

The chronology of MAYA-22-7 was determined with a U-Th age model constructed from five U-Th ages drilled from the central working slab (Fig. 1). Sample preparation and analytical chemistry for dating were conducted at the Department of Earth Sciences, University of Oxford. Samples were dissolved in distilled concentrated nitric acid and spiked with a mixed $^{229}$Th-$^{236}$U solution. U and Th were separated using column chemistry, following procedures adapted from Edwards et al. (1987). U and Th isotopes were measured using a Nu Plasma II Multi-ion-counting, Multi-collector Inductively Coupled Plasma Mass Spectrometer (MC-ICP-MS), following procedures adapted from Hoffman (2008). Ages were calculated iteratively using the $^{230}$Th/$^{238}$U age equation (Kaufman and Broecker, 1965), derived in Richards and Dorale (2003). The $^{234}$U and $^{230}$Th half-lives used for age calculations are those reported in Cheng et al. (2013). Age uncertainty was calculated using a Monte Carlo method, which accounts for instrument measurement, chemical blank, and initial detrital $^{230}$Th/$^{232}$Th activity ratio uncertainty. The initial detrital $^{230}$Th/$^{232}$Th activity ratio is set as a uniform distribution of values between 0.1 and 2.0, which encompasses the bulk earth value ($^{230}$Th/$^{232}$Th activity = 0.82) and the median detritus value determined from a collection of speleothem studies ($^{230}$Th/$^{232}$Th activity = 1.5) (Hellstrom 2006). All U-Th ages are in sequential order, with the corrected youngest date (9.3 mm from stalagmite top) 2001 ± 11 (2σ) CE.

The interpolated age model for the stalagmite central growth axis was produced using the OxCal version 4.4 Poisson-process Deposition model, with the stalagmite vertical growth rate constrained using model inputs $k_0 = 0.1$ mm$^{-1}$ and $\log_{10}(k/k_0) = U(-2, 2)$ (Ramsey, 2008, 2009; Ramsey and Lee, 2013). Because sample pits in this study were drilled off the central axis, visible laminae were tracked to connect the sample pit locations to the central growth axis to determine the mean age range and 95% confidence age range of the sample pits. The OxCal Deposition model input code, an age-depth plot (Fig.


S2), U-Th isotope activity ratios and calculated age of U-Th samples (Table S2), as well as ages and their respective errors of the biomarker samples (Table S3) are found in the supplement.

## 2.2 PAH and MA extraction

The sample preparation procedure is illustrated in Fig. S3. For PAH extraction, powdered speleothem samples were weighed into baked-out (>8 h at 450 °C) 20 mL glass vials and spiked with 100 µL of a solution of 100 ng mL$^{-1}$ naphthalene-$d_8$, acenaphthene-$d_{10}$, and phenanthrene-$d_{10}$ in ethyl acetate (EA). Subsequently, each sample was extracted twice with 5 mL g$^{-1}$ dichloromethane (DCM) in an ultrasonic bath for 45 min. After sonication, samples were allowed to stand for 10-20 minutes to improve phase separation. The supernatant was removed, filtered through a 1 µm glass fibre filter (Macherey Nagel), and

loaded onto the preconditioned (3 mL each of EA and DCM) SPE cartridges. The flow-through was collected in baked-out 20 mL glass vials with tapered bottoms. The cartridge was rinsed with 3 mL each of DCM and EA and dried by blowing air through the cartridge. Subsequently, the solution was evaporated under a gentle stream of nitrogen at 30 °C to a volume of approximately 500 µL. The walls of the vial were washed with 3 mL EA, the solution was once again evaporated to approximately 500 µL, and the walls were washed again with another portion of 3 mL EA. Then the sample solution was

evaporated to approximately 200 µL, the walls were washed a last time with 500 µL EA, and the solution was finally evaporated to a volume of ~100 µL. This solution was transferred into a baked out 250 µL vial. The evaporation vial was washed with 100 µL EA and the solution was added to the sample. To enable volume calculation and corrections during measurement, 2 µL of a solution of *p*-terphenyl (7.5 µg mL$^{-1}$ in EA) were added. The sample solution was stored in the freezer at -25 °C.

The extracted speleothem carbonate powder was dried in an oven at 50 °C overnight and then spiked with 100 µL of a solution of 100 ng mL$^{-1}$ $^{13}C_6$-levoglucosan in acetonitrile (ACN). The extraction procedure is described in Homann et al. (2022). In brief, two 45 min ultrasonic extractions were performed using 5 mL g$^{-1}$ methanol (MeOH) as an extraction agent. The supernatant was filtered with a 1 µm glass fibre fabric filter and evaporated to dryness under a gentle stream of nitrogen at 30 °C. The residue was redissolved in ACN/$H_2O$ (95:5) and filtered through a 0.2 µm PA-filter (Altman-Analytik) prior to

storage at -25 °C.

## 2.3 PAH analysis

The analysis was performed using a Thermo Fisher Scientific Orbitrap Exploris GC system. Analytes were separated on a TG-5-SILMS column (30 m, 0.25 mm inner diameter, 0.25 µm film thickness, Thermo Fisher Scientific). A volume of 1 µL was injected in splitless mode at an injector temperature of 320 °C, a transfer-line temperature of 320 °C, and an oven temperature

of 50 °C. The carrier-gas (helium, 5.0, Nippon Gases) flow was set to 1 mL min$^{-1}$. The initial temperature was held for 2 min, then increased to 160 °C at a rate of 10 °C min$^{-1}$ (1 min hold), then to 270 °C at 3 °C min$^{-1}$, then to 300 °C at 30 °C min$^{-1}$ (5 min hold), and finally to 320 °C at 30 °C min$^{-1}$ (2.7 min hold). The injector was cleaned after each injection for 5 min at 330 °C with a flow rate of 150 mL min$^{-1}$. The mass spectrometer was operated in positive electron ionisation mode (EI+) using



selected ion monitoring mode (SIM). The SIM *m/z* ratios and retention times of the respective PAHs are found in Table 1.

Analytes were quantified via the ratio of the peak area of the analyte to the peak area of the internal standard.

**Table 1: Trivial name, acronym, sum formula, retention time, *m/z* value, limit of detection, repeatability, and accuracy of all d-PAHs, internal standards, and PAHs determined by the GC-MS method.**

| | Analyte | Acronym | Sum formula | $t_R$ (min) | *m/z* [M]$^{·+}$ | LOD (ng) | Repeatability (%) | Accuracy (%) |
|---|---|---|---|---|---|---|---|---|
| Spiked d-PAHs and internal standard | Napthalene-d$_8$ | d-NAP | $C_{10}D_8$ | 9.49 | 136.1128 | 0.97 | 4.5 | 8.3 |
| | Acenaphthene-d$_{10}$ | d-ACE | $C_{12}D_{10}$ | 13.67 | 164.1410$^†$ | 1.37 | 4.6 | 1.5 |
| | Phenanthrene-d$_{10}$ | d-PHE | $C_{14}D_{10}$ | 19.40 | 188.1410 | 1.29 | 11.4 | 2.6 |
| | p-Terphenyl | PTP | $C_{18}H_{14}$ | 29.90 | 230.1096 | / | 6.6 | / |
| Investigated PAHs | Naphthalene | NAP | $C_{10}H_8$ | 9.53 | 128.0626 | 1.41 | 5.7 | 0.6 |
| | Acenaphthylene | ACY | $C_{12}H_8$ | 13.28 | 152.0626 | 0.07 | 7.0 | 0.2 |
| | Acenaphthene | ACE | $C_{12}H_{10}$ | 13.77 | 154.0783 | 0.14 | 5.1 | 0.3 |
| | Fluorene | FLN | $C_{13}H_{10}$ | 15.43 | 166.0783 | 0.17 | 25.6 | 0.1 |
| | Phenanthrene | PHE | $C_{14}H_{10}$ | 19.50 | 178.0783 | 2.07 | 17.6 | 4.8 |
| | Anthracene | ANT | $C_{14}H_{10}$ | 19.77 | 178.0783 | 0.04 | 3.3 | 0.1 |
| | Fluoranthene | FLT | $C_{16}H_{10}$ | 26.45 | 202.0783 | 0.67 | 6.2 | 1.0 |
| | Pyrene | PYR | $C_{16}H_{16}$ | 27.81 | 202.0777$^†$ | 0.34 | 2.2 | 0.5 |
| | Retene | RET | $C_{18}H_{18}$ | 30.48 | 234.1409 | 0.36 | 19.4 | 0.7 |
| | Benzo(a)anthracene | BAA | $C_{18}H_{12}$ | 36.53 | 228.0939 | 1.09 | 3.7 | 1.2 |
| | Chrysene | CHR | $C_{18}H_{12}$ | 36.75 | 228.0939 | 1.14 | 8.9 | 0.9 |
| | Benzo(b)fluoranthene | BBF | $C_{20}H_{12}$ | 44.02 | 252.0939 | 1.30 | 8.7 | 0.2 |
| | Benzo(a)pyrene | BAP | $C_{20}H_{14}$ | 45.99 | 252.0939$^†$ | 0.55 | 9.6 | 1.5 |
| | Indeno(1,2,3-c,d)pyrene | INP | $C_{22}H_{14}$ | 52.00 | 276.0939$^†$ | 0.51 | 10.2 | 1.1 |
| | Dibenz(a,h)anthracene | DBA | $C_{22}H_{14}$ | 52.18 | 278.1096 | NA | 12.0 | 1.0 |
| | Benzo(g,h,i)perylene | DPE | $C_{22}H_{12}$ | 52.73 | 276.0939 | 0.47 | 12.4 | 1.5 |

$^†$: Detection of fragment.

**2.4 MA analysis**

Analysis of MAs was carried out on a Dionex UltiMate 3000 ultrahigh-performance liquid chromatography system (UHPLC), coupled to a heated electrospray ionisation source (HESI) and a Q Exactive Orbitrap high-resolution mass spectrometer (HRMS) (all by Thermo Fisher Scientific) equipped with an iHILIC-Fusion column, 100 mm x 2.1 mm with 1.8 μm particle size (Hilicon). The injection volume was 10 μL. A H$_2$O/ACN isocratic program was applied at a flow of 0.3 mL min$^{-1}$ with a





run time of 5 min. The eluent composition was 97 % eluent B (100 % ACN) and 3 % eluent A (consisting of 5 mmol L$^{-1}$ ammonium formate in H$_2$O). To improve ionisation, a post-column flow of 50 mmol L$^{-1}$ NH$_4$OH in MeOH was applied with a flow rate of 0.1 mL min$^{-1}$. The HESI source was operated in negative mode to form deprotonated molecular ions [M-H]$^-$. The HESI probe was heated to 150 °C, the capillary temperature was set to 350 °C, and the spray voltage was -4.0 kV. The sheath gas pressure was 60 psi and the auxiliary gas pressure was 20 psi. The mass spectrometer was operated in full scan

mode with a resolution of 70,000 and a scan range of *m/z* 80–500. During the expected retention times (Table 2), the full scan mode was alternated with a targeted MS$^2$ mode with a resolution of 17,500. For the MS$^2$ mode (i.e. parallel reaction monitoring mode in the software Xcalibur, provided by Thermo Fisher Scientific), higher-energy collisional dissociation (HCD) was used with 35% normalised collision energy (NCE).

**Table 2: Trivial name, acronym, sum formula, retention time, *m/z* value, limit of detection, repeatability, and accuracy of $^{13}$C-MAs and MAs determined in the MA LC-MS method.**

| Analyte | Acronym | Sum formula | $t_R$ (min) | *m/z* [M-H]$^-$ | LOD (ng) | Repeatability (%) | Accuracy (%) |
|---|---|---|---|---|---|---|---|
| $^{13}$C$_6$-levoglucosan | $^{13}$C-LEV | $^{13}$C$_6$H$_{10}$O$_5$ | 3.28 | 167.0657 | NA | 11.5 | 0.1 |
| Mannosan | MAN | C$_6$H$_{10}$O$_5$ | 2.65 | 161.0455<br>129.0193[†] | 0.05 | 4.4 | 0.3 |
| Galactosan | GAL | C$_6$H$_{10}$O$_5$ | 2.77 | 161.0455<br>113.0245[†] | 0.01 | 11.0 | 0.2 |
| Levoglucosan | LEV | C$_6$H$_{10}$O$_5$ | 3.30 | 161.0455<br>113.0245[†]<br>101.0244[†] | 0.09 | 6.4 | 0.1 |

[†]: Fragments resulting from targeted MS$^2$.

## 3 Results and discussion

### 3.1 Method validation

Detection limits (LODs) were calculated using Eq. (1), where B is the blank and SD$_B$ the corresponding standard deviation (Otto 2014).

$$LOD = B + 3\,SD_B \,, \tag{1}$$

Recovery was calculated by spiking samples with known amounts of naphthalene-d$_8$ (42-102 %), acenaphthene-d$_{10}$ (41-120 %), phenanthrene-d$_{10}$ (46-137 %), and $^{13}$C$_6$-levoglucosan (62-83 %). The comparatively low recovery rates were likely a

consequence of volatilisation during the evaporation step. A comprehensive overview of all recoveries is found in Table S3. A 25 ng mL$^{-1}$ (PAHs) or a 2.5 ng mL$^{-1}$ (MAs) standard was used to calculate the accuracy of the method (percent relative





deviation with respect to the standard concentration). Repeatability was calculated as the standard deviation of nine measurements of 25 ng mL$^{-1}$ (PAHs) or 2.5 ng mL$^{-1}$ (MAs) standards. Results are summarised in Tables 1 and 2.

### 3.2 Speleothem MAs and PAHs

Levoglucosan (LEV) and Σ15 (sum of non-alkylated PAHs) concentrations are presented in Fig. 2. Concentrations of individual MAs and PAHs are found in Table S4, whereas calculated sums and ratios are shown in Table 3. The LEV and Σ15 concentrations range from 0.6 to 5.7 ng g$^{-1}$ and 3.8 to 16.9 ng g$^{-1}$, respectively. The Σ15 concentrations are greater than values presented by Argiriadis et al. (2019) but lower than those reported by Perrette et al. (2008). Σ15 concentrations and to a lesser extent, LEV concentrations, track the colour (grey value) of the stalagmite (Fig. 2Figure 2), suggesting a link between grey

scale value and fire activity (soot).

**Table 3: Calculated sums and diagnostic ratios of the 10 samples from stalagmite MAYA-22-7 from Cenote Ch'en Mul, Mayapan.**

| Age (Year CE)* | 2010-2015 | 2000-2010 | 1995-2000 | 1945-1985 | 1945-1965 | 1920-1935 | 1890-1905 | 1855-1875 | 1760-1790 | 1640-1650 |
|---|---|---|---|---|---|---|---|---|---|---|
| Sample # | 4 | 3 | 5 | 2 | 6 | 8 | 9 | 1 | 10 | 11 |
| Σ15† | 6.44 | 6.2 | 9.95 | 14.7 | 15.4 | 3.8 | 11.9 | 6.7 | 10.3 | 5.4 |
| LMW‡ | 6 | 5.6 | 9 | 13.8 | 14.5 | 3.7 | 11.3 | 6 | 9.7 | 5.04 |
| HMW◊ | 0.4 | 0.5 | 0.9 | 0.9 | 0.9 | 0.02 | 0.6 | 0.7 | 0.5 | 0.3 |
| LMW/HMW | 13.6 | 10.4 | 10.33 | 14.9 | 15.8 | 175.5 | 19.3 | 8.5 | 18.9 | 14.7 |
| PHE/ANT | 74.7 | 4 | 43 | 3.4 | NA | NA | NA | 4.8 | NA | NA |
| ANT/(ANT+PHE) | 0.01 | 0.2 | 0.02 | 0.2 | NA | NA | NA | 0.2 | NA | NA |
| RET/(RET+PHE+ANT) | NA | NA | 0.16 | NA | NA | NA | 0.1 | NA | NA | NA |
| LEV/(LEV+Σ15) | 0.5 | 0.2 | 0.4 | 0.4 | 0.5 | 0.6 | 0.3 | 0.3 | 0.4 | 0.5 |
| LEV/MAN | 5.9 | 9.7 | 8 | 9.5 | 18.8 | 7.4 | NA | 9.1 | 8.8 | 6 |
| LEV/(MAN+GAL) | 3.3 | 4.7 | 4.6 | 4.8 | 5.9 | 3.7 | 18.6 | 5.4 | 4.7 | 3.1 |

*: Start and end dates have errors of ± 6-20 years (95 % confidence interval) †: Sum of non-alkylated PAHs; ‡: Sum of two and three-ring PAHs; ◊: Sum of four and five-ring PAHs. Errors of sums and diagnostic ratios are reported in Table S5.




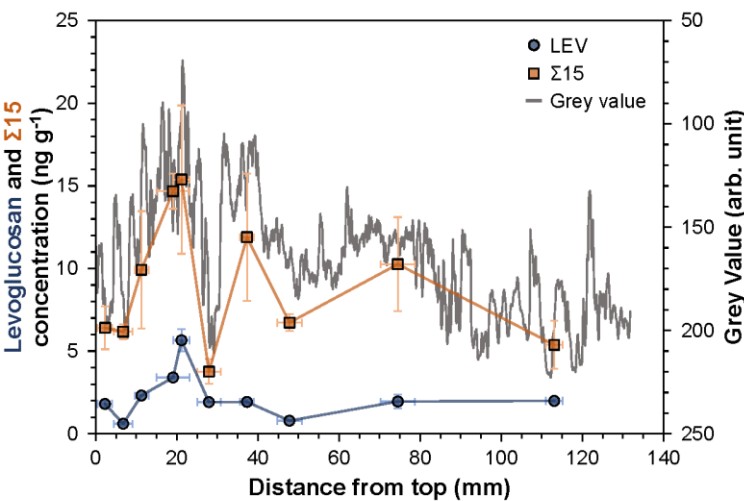

**Figure 2: Concentrations of levoglucosan (dark blue) and Σ15 (sum of non-alkylated PAHs; orange) in the 10 analysed samples from stalagmite MAYA-22-7. Symbols are centred on the middle of their respective sample pit (Fig. 1). Error bars in the x-axis reflect the lengths of the sample pits. Error bars on the y-axis represent one standard deviation and may be smaller than the symbols. The grey line shows the grey value of stalagmite MAYA-22-7, measured along the growth axis.**

Overall, a strong correlation ($r = 0.75$, $p < 0.05$) between LEV and Σ15 is observed (Figure 3 3). This correlation changes only slightly ($r=0.76$, $p < 0.05$) when solely low-molecular-weight PAHs (LMW, sum of two- and three-ring PAHs) are considered. This is consistent with the findings of Battistel et al. (2017) and suggests a common origin for LEV and smaller PAHs. The three most abundant PAHs (phenanthrene (PHE), naphthalene (NAP), and fluorene (FLN)) constitute an average of 82 ±7 % of Σ15 in the analysed samples. However, PHE, the most abundant PAH, does not show a significant correlation with LEV ($r=0.60$, $p < 0.1$, Figure 3, green circles).

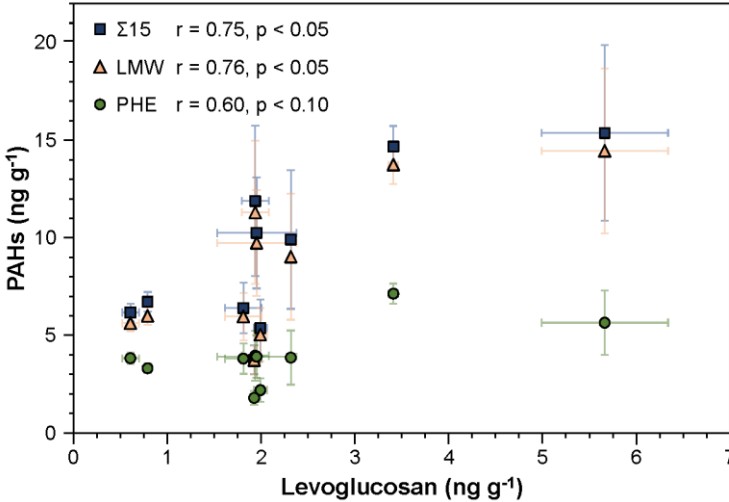





**Figure 3: Correlation of levoglucosan and PAHs in the 10 samples analysed from stalagmite MAYA-22-7. Blue squares: Σ15 (sum of non-alkylated PAHs); beige triangles: LMW (sum of two and three-ring PAHs); green circles: PHE. Error bars represent one**
**standard deviation and may be smaller than the symbols.**

Most high molecular weight PAHs (HMW, four- and five-ring PAHs) are below LOD in the analysed sample set. This is consistent with the results of Argiriadis et al. (2019) and Perrette et al. (2008) and is likely a consequence of a filtering by the overlying soil and epikarst, as discussed by Perrette et al. (2013).

**Table 4: Typically reported values of diagnostic ratios of PAHs and MAs, and respective interpretations.**

| Ratio | Values | Interpretation | Reference | Medium applied to |
|---|---|---|---|---|
| LMW/HMW | < 1 | Pyrogenic | Zhang et al. (2008) | Water |
| | > 1 | Petrogenic | | |
| PHE/ANT | < 10 | Pyrogenic | Zhang et al. (2008) | Water |
| | > 15 | Petrogenic | | |
| ANT/(PHE+ANT) | < 0.1 | Petrogenic | Yunker et al. (2002a) | River sediment |
| | > 0.1 | Pyrogenic | | |
| RET/(RET+CHR) | 0.15-0.50 | Petrol combustion | Yan et al. (2005) | Lake sediment |
| | 0.30-0.45 | Coal combustion | | |
| | 0.83-0.96 | Softwood combustion | | |
| RET/(RET+PHE+ANT) | > 0.1 | Gymnosperm combustion | Karp et al. (2020) | Aerosols |
| | < 0.1 | Angiosperm combustion | | |
| LEV/(LEV+Σ15)[†] | < 0.5 | High-intensity fires | Ruan et al. (2020) | Marine Sediment |
| | 0.5 | Boundary | | |
| | > 0.5 | Low-intensity fires | | |
| LEV/MAN | 0.5-14 | Softwoods | Fabbri et al. (2009) and | Aerosols; Chars |
| | 3-32 | Hardwoods | references therein; Kuo | |
| | 4-55 | Grasses | et al. (2011) | |
| | 30-90 | Lignite | | |
| LEV/(MAN+GAL) | 0.5-6 | Softwoods | Fabbri et al. (2009) and | Aerosols; Chars |
| | 1.5-10 | Hardwoods | references therein; Kuo | |
| | 4-55 | Grasses | et al. (2011) | |
| | 30-90 | Lignite | | |

[†]: LEV and Σ15 are normalised to the highest value of the series.

The lack of HMW PAHs means that some PAH ratios normally used to characterise PAH profiles (Table 4) may not apply to speleothems. An example of such a ratio is LMW/HMW; for directly deposited samples such as aerosols or sediments, a ratio
> 1 indicates a petrogenic origin of PAHs (Soclo et al. 2000; Zhang et al. 2008). The artificially lowered HMW levels observed





in speleothems results in very high LMW/HMW ratios (Table 3), rendering this ratio insensitive to the origin of the PAHs. More promising diagnostic metrics in speleothem analysis could be individual PAHs. For this purpose, we studied the ratios of two or more compounds with similar molecular formulas but different molecular structures, assuming that the compounds have comparable physicochemical properties (i.e. volatility, solubility, atmospheric lifetime, adsorption, and transport

mechanisms) but different predominant origins (Yunker et al. 2002b). At high temperatures, a less stable, "kinetic" isomer is formed, which has more aromatic rings and less alkyl substitutes compared to the product at lower combustion temperatures or of petrogenic origin (Han et al. 2020; McGrath et al. 2003; Tobiszewski und Namieśnik 2012). Phenanthrene (PHE) and anthracene (ANT) form one pair, where PHE is the thermodynamic and ANT the kinetic isomer (Soclo et al. 2000). The resulting PHE/ANT and ANT/(ANT+PHE) ratios can also be used to distinguish between pyrogenic (PHE/ANT < 10;

ANT/(ANT+PHE) > 0.1) and petrogenic (> 10; < 0.1) origins of the PAHs and appear to apply to our stalagmite samples. The PHE-ANT ratios of samples #1-3 indicate pyrogenic origin; for the remainder of the samples, ANT is very close to or below the LOD. Therefore, we cannot conclude with certainty a petrogenic origin of the PAHs in samples #4-11.

Retene (RET) is an alkylated PAH and a molecular marker of gymnosperm combustion (Muri et al. 2003; Ramdahl 1983; Wakeham et al. 1980). Unfortunately, RET was below the LOD for all but two samples; nevertheless, RET might provide a

promising marker of fire dynamics in speleothems if it is present in sufficient concentrations. RET is also used in various diagnostic ratios. For instance, in combination with chrysene (CHR), where RET/(RET + CHR) helps to determine fuel type (Yan et al. 2005). Because CHR is one of the HMW PAHs, it is below the LOD for all samples in this study, but this ratio could potentially be utilised in other samples. In addition, Karp et al. (2020) suggested the use of a ratio that includes three 3-ring PAHs, RET, PHE, and ANT, where RET/(RET+PHE+ANT) is used to distinguish between gymnosperm (softwood,

> 0.1) and angiosperm (hardwood, < 0.1) combustion. As mentioned above, in our sample set, RET is above the LOD in only two samples. For one of these (1995-2000 CE) the ratio points towards gymnosperm fuel whereas the result for the other sample (1890-1905 CE) falls right on the boundary between angiosperm and gymnosperm fuel. This ratio could be particularly useful in combination with the MAs, which can also be used to determine fuel type.





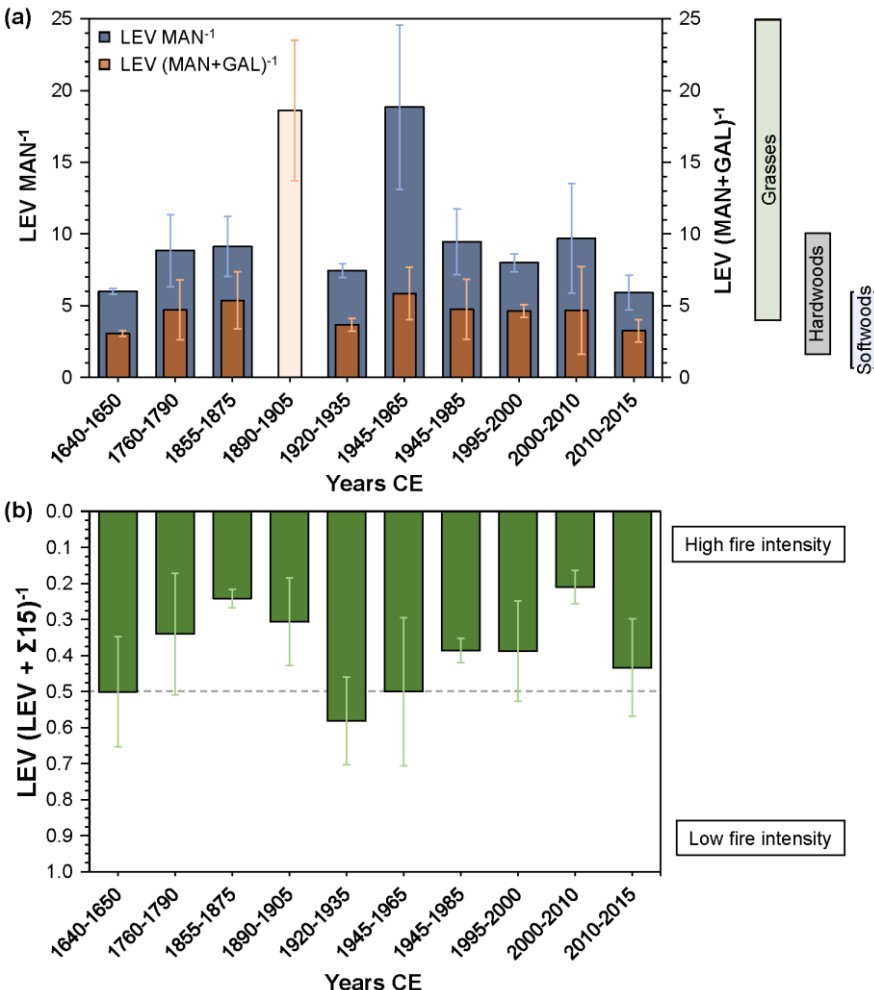

**Figure 4: (a) LEV MAN⁻¹ (dark blue) and LEV (MAN + GAL)⁻¹ (orange) ratios of the 10 samples analysed from stalagmite MAYA-22-7. Coloured bars on the right indicate the burned fuel based on the LEV (MAN + GAL)⁻¹ ratio according to Fabbri et al. (2009) and Kuo et al. (2011). Green: grasses; grey: hardwoods; blue: softwoods. Age range listed is the OxCal modelled mean age at the top and bottom of each sample pit. Start and end dates have errors of ± 6-20 years (95 % confidence interval; see Table S2). Error bars represent one standard deviation. (b) LEV (LEV + Σ15)⁻¹ ratios of the 10 samples analysed. The dashed line represents the**
**boundary between MA-dominated fire regimes (low intensity) and PAH-dominated fire regimes (high intensity). Age range listed is the OxCal modelled mean age at the top and bottom of each sample pit. Start and end dates have errors of ± 6-20 years (95 % confidence interval; see Table S2). Error bars represent one standard deviation.**

MA ratios, LEV MAN⁻¹ and LEV (MAN + GAL)⁻¹, respectively, have been shown to capture the type of biomass burned (Fabbri et al. 2009 and references therein; Kuo et al. 2011). According to Fabbri et al. (2009), the use of LEV (MAN + GAL)⁻¹

is even more sensitive and discriminating. The ratios in our study are shown in Figure 4Fig. 4 (a) and indicate combustion of predominantly mixed hardwoods and softwoods. Kuo et al. (2011), however, showed that high combustion temperatures may shift the MA ratios to higher values because hemicellulose is less thermally stable than cellulose. This means that MAN and GAL are not emitted at higher temperatures, whereas LEV can still be formed. This could be the case for our sample from



1890-1905 CE (light orange in Fig. 4 (a)), in which no MAN could be detected and the resulting LEV $(MAN + GAL)^{-1}$ is very

high. Therefore, it might be useful to combine the MA ratios with an indicator of fire intensity. Ruan et al. (2020) suggested the use of LEV $(LEV + \Sigma15)^{-1}$, with both LEV and $\Sigma15$ normalised to the highest value in the series. Values > 0.5 indicate a predominance of PAHs and high fire intensity, whereas values < 0.5 indicate low intensity fires with dominant MA emission. Fig. 4 (b) shows two distinct peaks in fire intensity (i.e. low LEV contribution) in the 1855-1875 CE and 2000-2010 CE samples. The 1890-1905 CE sample mentioned above also shows high intensity fires, supporting the interpretation that the

LEV $(MAN + GAL)^{-1}$ ratio may have been artificially increased. The 1920-1935 CE sample is the only one that clearly indicates low fire intensity, whereas the remaining samples fall into the high fire intensity regime. We note however, the relatively large analytical uncertainties make a definitive interpretation difficult.

### 3.3 Lev and PAHs as indicators of fire activity at Cenote Ch'en Mul

We applied our results to infer past fire activity in and around Cenote Ch'en Mul. Mayapan was the last capital of the Late

Postclassic Maya period and was abandoned in the mid-15th century (Kennett et al. 2022; Milbrath und Peraza Lope 2003). Its ruins, including the round temple and the Kukulcan temple, were first reported to the English speaking world in 1841 by John Stephens. Although preliminary excavations were conducted in the early 20th century, major archaeological work only took place in the 1950s under the auspices of the Carnegie Institution (Milbrath und Peraza Lope 2003). These archaeological activities are relevant to the present study, because human-induced fires related to the excavations (land-clearing, use of

kerosene lamps or open fires, etc.) produced LEV and PAH that should be recorded in our dataset. Activities like water collection or ritual incense burning have a very long history in Yucatan and are still conducted to this day. Any open flame combustion thus adds to the overall PAH and MA load in the cave, and potentially in speleothems. The signal observed in a stalagmite represents the integral of all accumulated PAHs and MAs, and in-depth research is required to differentiate the relative contribution of each pathway.

Stalagmite MAYA-22-7 grew fast (growth rate of 200-300 µm/year before the dark layer at a distance of 15-23 mm from the top, and 500-600 µm/year after that dark layer) and covers only the last ~400 years, i.e. the period after the site was abandoned by the Postclassic Maya. The fire history inferred from the stalagmite is likely related to both natural fires in the low-stature, thorny, deciduous tropical forest that charaterised the area, and land-clearing activities that preceeded archaeological excavation at the site. Early and middle 20th century archaeological efforts likely used kerosene lamps for work in Cenote

Ch'en Mul. Parts of the Mayapan site might have been cleared of vegetation using fire, although it is difficult to find evidence for this practice in archeological field reports. The Carnegie excavations might have used both kerosene and battery powered lights. Either way, it is plausible that LEV and PAHs from human induced fires might be recorded in the stalagmite. This is in accordance with our findings, in that the samples covering the period 1945-1985 show the highest LEV and PAH concentrations in the whole sample set. Finding elevated PAH and MA concentrations in speleothem carbonate that dates to a

time of higher human-induced fire activity is encouraging and suggests that speleothems are sensitive recorders of combustion products. Further work should be directed towards i) deciphering transport mechanisms and potential scavenging effects in the



soil and epikarst, ii) detailed understanding of incorporation pathways into carbonate, and iii) refinement of methods to delineate dripwater- and aerosol-derived fire proxies.

**4 Conclusions and outlook**

We used a sequential extraction preparation method for analysis of 16 PAHs and three MAs in samples from a stalagmite collected at the Postclassic Maya archaeological site of Mayapan, Yucatán, Mexico. Sample preparation involved solid-liquid extraction of pulverised carbonate samples. To our knowledge, this is one of only a few methods for analysis of PAHs in speleothems and the only one that also enables analysis of MAs. The new method requires relatively small sample amounts (0.5-1.0 g) and simple instrumentation, i.e. an ultrasonic bath and a sample concentrator. PAHs and MAs were measured by
GC-MS and LC-HILIC-MS, respectively.

We applied the method to 10 samples from the MAYA-22-7 stalagmite, collected at Cenote Ch'en Mul, directly below the site of Mayapan. We used concentrations and selected ratios of PAHs and MAs described in the literature to explore fire history in this area of the Yucatán Peninsula. We found a positive correlation between the major MA, levoglucosan, and non-alkylated PAHs ($\Sigma15$), or two- and three-ring PAHs (LMW), (r = 0.75, p < 0.05, and r = 0.76, p < 0.05, respectively). Not all diagnostic
PAH ratios were applicable, but the ratios combining PAHs with MAs appear to have promise for discriminating among different fire regimes and inferring the source of burned fuel. Furthermore, retene (RET) can be used as a molecular marker for gymnosperm combustion, if present in high enough concentrations.

Our analyses suggest that within the analysed time series, the period 1945-1985 stands out in terms of high fire activity. We tentatively interpret this result to indicate clearing of the site prior to and during mid-20th-century archaeological excavation.
A longer record of the two biochemical proxies from older stalagmites from Che'Mul Cave could document the history of cave use by the Maya for ceremonial and practical purposes - burning undoubtedly occurred within the cave (e.g. torches for lighting).

For future work, the two biochemical proxies studied here should be combined with other, established proxies such as carbon and oxygen isotope ratios as well as trace elements described by Campbell et al. (2023) to gain further insights into the interplay
among fire, hydrology, vegetation, and human activities. In addition, monitoring rainfall and dripwater at the sample site would be useful to track changes in PAH and MA patterns and to gain a deeper understanding of site-specific transport mechanisms. It would also be beneficial to apply our method to other speleothems from the same cave and other sites, to test whether our results are generally applicable. Finally, other source-specific PAH species could be added to enhance the interpretation of the PAH patterns. Potentially suitable PAHs include 1,2- and 1,7-dimethylphenanthrene (1,2- and 1,7-DMP). Kappenberg et al.
(2019) suggested that their ratio could be used as an alternative to distinguish among different types of burned biomass.

**Data availability**. All data from our study are presented numerically in the paper and in the Supplement.



**Author contributions**. JH and TH designed the study. JH and NK conducted the method development, sample preparation,
and data analysis for PAHs and MAs. CL and MB enabled sampling in Cenote Ch'en Mul. DJ, OK, DH, and SB collected
stalagmite MAYA-22-7. OK and SB surveyed Cenote Ch'en Mul and prepared the cave map. SC and DJ subsampled the
stalagmite for analysis and constructed the age model. All co-authors participated in discussions, interpretation of the data, and
writing of the manuscript.

**Competing interests**. The contact author has declared that none of the authors has any competing interests.

**Acknowledgements**. The authors thank Susan Milbrath and Marylin Masson for insights on the history of the archaeological
excavations. JH gratefully acknowledges financial support from the Max Planck Graduate Center Mainz. Some of the
analytical developments included here were developed with financial support from DFG Project HO 1748/20-1 (Organic trace
analysis of atmospheric marker substances in ice cores). DH acknowledges financial support from the Leverhulme Trust Award
RPG-2019-228.

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
