# Peer review of "Past fire dynamics inferred from polycyclic aromatic hydrocarbons and monosaccharide anhydrides in a stalagmite from the archaeological site of Mayapan, Mexico"

_Biogeosciences, 2023_

## Author Response (AR1)

**Public Justification**

Thanks for submitting your interesting manuscript to Biogeosciences and for responding to the comments/suggestions from the reviewers.

I agree with the reviewers that only minor revision is needed. You have responded appropriately to all comments and it is clear how you plan to address these in the next version. Therefore, I invite you to submit your revised manuscript that includes the changes as per your responses.

Reply:

Thank you for the positive response and support of our work.

As additional comment and regarding the question about retene and potential gymnosperm plant input, I wonder if you have also investigated other compounds in your lipid extracts? As part of your method, you should be able to do a general biomarker screening (in selected samples at least), which would allow you to check for the presence of other gymnosperm-derived biomarkers in addition to angiosperm biomarkers? In addition to that or alternatively, a short description of dominant vegetation at the study site may be useful.

In this approach, you may also find other biomass burning indicators, such as those summarised in Simoneit (2002) for instance (https://doi.org/10.1016/S0883-2927(01)00061-0), though PAHs and anhydrosugars are certainly the most reliable organic indicators of past fires.

Reply:

Thank you for your suggestions. Concerning the vegetation at the site we added the following note:

The vegetation at the study site is composed of a low-stature, thorny, deciduous tropical forest (lines 298-299).

We appreciate your input concerning additional specific biomass burning markers. We discuss this topic in more detail now in the Future Work Section of our manuscript (lines 342-344).

Thank you for these edits and I look forward to your revised manuscript.

**RC1**: 'Comment on bg-2023-63', Anonymous Referee #1, 10 May 2023

Comment**:** The paper "Past fire dynamics inferred from polycyclic aromatic hydrocarbons and monosaccharide anhydrides in a stalagmite from the archaeological site of Mayapan, Mexico" presents an interesting comparison between different biomarkers of past fire activity as records in a speleothem. Such comparisons are necessary in order to better understand the strengths and weaknesses of biomarkers, and especially such differences as recorded in a matrix in which they are not often analyzed. However, the concentrations of these biomarkers are often below the limit of detection (LOD), thereby making the comparisons less useful. The values below the LOD also influence their ability to be used in ratios that help determine the type of past burned vegetation. It is unclear if these low values are due to the matrix, the location, or the sampling procedure. This paper still deserves to be published, but the limited values should be mentioned in the introduction.

Reply:

We thank Referee #1 for the thorough and positive review, appreciation of our work, and recommendation to publish it.

We now acknowledge the limited number of biomarker values presented in the paper, and elaborate on this point in the revised manuscript, in the Introduction ("Even though some of the biomarkers were below the limit of detection (LOD), the analysed stalagmite samples provided valuable hints about past fire dynamics") and Conclusion ("However, it has to be noted that the HMW PAHs were often below LOD, probably because of a filtering effect of the overlying soil. This limited our ability to fully exploit the data set to interpret past fire dynamics at Mayapan.")

Please also address the following points:

Minor:

Line 57: Please explain what you mean by "cave-internal sources".

Reply:

"Cave-internal sources" refers to combustion sources within the cave, caused by humans (e.g. campfires, torches, candles, or petroleum lamps).

Line 56 was modified to: "[...] by cave-internal sources (e.g. campfires, torches, candles, or petroleum lamps)."

Lines 165-170: What are the benefits of running in the full scan mode rather than targeting the known m/z? If the retention times are known, then why not just use the targeted MS[2] mode? Is anything gained from using the full scan mode?

Reply:

We used a full scan because it offers the possibility for a non-target approach, even if that was not applied in this study, and it is less sensitive to shifts in the retention time, which often occur in matrix-heavy samples.

Section 3.1: At what point in the study was the recovery rates tested? As the existing method resulted is relatively low recovery rates, why was the method not adapted accordingly to minimize volatilization and therefore to improve recovery? Were other solvents tested that would minimize volatilization? Other solvents may also influence other aspects of the recovery, but it is surprising that these solvents were not systematically tested and represented in this study as this work is a "proof-of-concept" work.

Reply:

The method we used was adapted from an existing method for the extraction of PAHs from filter samples (DOI: 10.5194/acp-18-13495-2018), consequently we did not perform a systematic solvent screening. We used dichloromethane (DCM) as the extraction agent because we knew from previous studies (DOI: 10.5194/egusphere-egu21-1065) that DCM does not impede the subsequent levoglucosan extraction. We have now included this information in the main text.

We agree that a wider screening approach should be mentioned in the Outlook section and we added a note to that effect in the Conclusions section ("For future work, the PAH extraction method should be improved to optimise recovery and reduce its variance. One possible approach is to perform a systematic solvent screening to improve the extraction efficiency.")

Line 214: Although Perette et al., 2013 discuss the possible effects of filtering by the overlying soil, this concept needs details within the submitted paper in order to be clear to the reader. The reader can hunt down Perette et al., 2013 and read the work, but ideally mentions of other literature should include sufficient detail that the readers understand why the work was cited. In this case, does the soil preferentially remove the high molecular weight PAHs? If so, what is the mechanism for this fractionation between the high molecular weight PAHs and the lower molecular weight PAHS? As noted in Lines 218-222 this filtration effect influences examining any ratios that incorporate low molecular weight PAHs.

Reply:

We are happy to clarify this in more detail in the manuscript and added the following explanation: "[Perette et al. 2013] found that HMW PAHs accumulate in soils and only LMW PAHs are transferred into the groundwater under steady state conditions."

Line 234-244: Is Retene influenced by the method (i.e. the potential loss during volatilization as mentioned in Section 3.1)? Could the method cause the fact that only two samples have quantifiable concentrations of Retene above the LOD? Could the filtration through the soil (Lines 218-222) also affect the Retene concentrations? With only two samples above the LOD, this paragraph is speculative, although the authors do acknowledge the limits of trying to interpret data with only two points.

Reply:

Retene has a very similar structure to phenanthrene (PHE), which shows the highest recoveries of the three $^{13}$C-PAHs investigated in our study. Perette et al. (2013) found PHE to be one of the PAHs that are transported into the groundwater. This led us to believe that the low retene concentrations were a consequence of a lack of gymnosperm vegetation and its combustion, rather than the methods used.

We implemented the following statement in the main text: "Literature suggests that gymnosperm vegetation is not common in the study area." (Contreras-Medina et al., 2007; Douglas et al., 2012; Etnoflora Yucatanense, 1994; Tellez et al., 2020)

Figure 4b: Why is the fire intensity data plotted with a reverse y-axis? The viewer does not gain any additional information from this flipped axis.

Reply:

We flipped the y-axis so that high fire intensities are at the top but altered the figure accordingly in the revised manuscript.

**Miscellaneous:**

Lines 40-41: The following existing sentence should be used to start a new paragraph: "The presence of specific biomarkers is indicative of the fuel source".

Line 48: Remove the word "for".

Table 2: "Common name" is a more accepted form than "trivial name".

Line 189: Only one "Figure 2" reference is necessary.

Reply:

We revised the manuscript to incorporate the 'miscellaneous' recommendations.

**RC2:** 'Comment on bg-2023-63', Anonymous Referee #2, 14 Jun 2023

Comment: This is an interesting paper, addressing the first joint analysis of different categories of fire biomarkers in speleothems. I support its publication with the relatively minor amendments detailed below.

Reply: We thank Referee #2 for the supportive review and enthusiastic recommendation to publish our work.

Comment: Method – I note that the speleothem samples were extracted by sonication of powdered samples, and not by dissolution. This means that inevitably there is only partial release of the organic compounds from the calcite. Given that, it would be good to include a note as to why this approach was chosen for the information of the non-chemist reader. In the discussion it would be also worth considering whether other approaches (e.g. microwave extraction) might be more effective than sonication.

Reply: In theory, extraction from a solution of the dissolved speleothem would be preferable, as all analytes would be available for extraction. However, our previous experimental work (DOI: 10.5194/egusphere-egu21-1065) found that isolation of MAs from solution is extremely difficult and thus inferior to extraction of powdered samples. In short, dissolution of the speleothem in acid results in a highly concentrated, polar solution. MAs are very polar molecules that preferentially reside in the acid phase rather than in the organic solvent used to extract them. Evaporation of said solution is also not advisable, as the MAs will adsorb to the salts that precipitate.

Our routine sample preparation procedure includes an additional third set of analytes (lignins) that are structurally similar to PAHs. After MA extraction, we dissolve the speleothem samples for lignin extraction. It would be very challenging to separate lignin and PAHs present in the same solution. This is another reason why we chose to extract the PAHs from the powdered samples prior to the MA extraction.

We discuss the method in the Future Work section in the revised manuscript.

Comment: I am concerned by the low recovery rates and also by the variability of the recovery rates between samples for each compound, particularly for PAHs – they seem to vary between around 40-50% to over 100%. That level of inconsistency is concerning for the reliability of the method, and as such deserves significantly more discussion, especially the impact on issues such as timeseries and source ratios. It should also be addressed in the future work section.

Reply: We agree that the original manuscript did not fully address the low recovery rates. They are indeed a challenge in application of our method and now, following your advice, we discuss possible remedies in the Future Work section. We added the following paragraph:

"The PAH extraction method should be improved to optimise recovery and reduce its variance. One possible approach is to perform a systematic solvent screening to improve the extraction efficiency. Alternatively, the extraction method could be modified, e.g. by utilisation of microwave extraction or accelerated solvent extraction. Another route is the addition of a keeper (e.g. nonane) during evaporation, to reduce losses to volatilisation, similar to the method described by Wietzoreck et al. (2022). To correct the data for recovery, additional deuterated PAH standards could be added."

Comment: It would be helpful, again for the non-chemist reader, to include an explanation of how the time-series data was corrected for the recovery %. Clarification as to how it is possible to recover more than 100% of the spike added would also be helpful.

Reply: The time series data were not corrected for recovery. To correct for recovery, a deuterated standard of all 16 PAHs would be needed because each component behaves differently. We acknowledge this issue in the Method Validation and Future Work sections of the revised manuscript.

Recovery of >100% might arise as a consequence of inaccurate volume correction, cross contamination, or carry over during the measurement, and is not uncommon (e.g. Argiriadis et al. 2019).

Comment: I agree with the authors that low HMW content could well be due to soil filtering – although this would not affect aerosol deposition from within the cave or via airflow. Some expansion of discussion on this point would be helpful, especially as it later is proposed that PAHs at the site could come from archaeological lamps and fires – lamps etc would presumably not have their signal filtered.

Reply: Referee #2 is correct in noting that the incorporation of airborne PAHs into speleothems is currently undocumented. To our knowledge, no existing literature covers this topic, and we can only speculate on what mechanisms were involved. As suggested, we elaborate on this in the Discussion section. We added the following paragraph:

"No literature addresses the inclusion of airborne PAHs into speleothems or whether fractionation between HMW and LMW PAHs might be expected. However, it is plausible that LMW PAHs are more readily dissolved in the water-film that covers actively growing speleothems, than are HMW PAHs. This is supported by the respective n-octanol/water partition coefficient ($K_{OW}$) and air/water partition coefficient ($K_{AW}$) values, according to which, for instance, NAP is 1-2 orders of magnitude more likely to enter the aqueous phase than BAA (Lu et al., 2008; Rayne and Forest, 2016). HMW PAHs adsorbed to particles could also be incorporated into the growing speleothem. However, Dickson et al. (2023) demonstrated that microscopic particles are situated on the flanks of a stalagmite rather than at the centre. As the samples analysed in this study were drilled along the growth axis, particulate PAHs would very likely not be encountered."

Comment: Line 205 mentions that PHE doesn't correlate with Lev – it would be good to suggest why this might be the case.

Reply: Thank you for pointing this out. We suspect that the observed discrepancy is caused by different sources of PHE and LEV. It is possible that PHE is generated by higher-intensity fires than required to produce LEV. We added a note about this in the revised manuscript (Lines 209-210).

Comment: Retene – what is the hypothesised reason for the absence of Retene in most samples – is it an environmental indicator (lack of gymnosperm combustion) or an artefact of preservation or analysis? How could future researchers resolve this? In the conclusion the impression is given that Retene as a gymnosperm marker is demonstrated by the findings in this paper – however, this does not appear to be the case, given it was below LOD in all but 2 samples? It's important to distinguish clearly between what these results demonstrate and what is being inferred from other literature.

Reply: In our reasoning we combine our results (low retene concentrations) and data from the literature (suggesting that gymnosperm vegetation is not common in the study area). We propose that negligible or below LOD retene concentrations reflect the lack of gymnosperm vegetation/combustion, rather than a methodological artifact, but we admit that the absence of evidence is not the same as evidence of absence. We formulated our conclusions more carefully to avoid misunderstanding.

We added a reference in lines 247-248.

Comment: Figures – given that section 3.3 is discussed by years, while figure 2 is presented as distance from top, I'd really like to see a new figure, probably in section 3.3 presenting the change in the key biomarker indicators (i.e. the ones the authors consider most valid at this site) through time. This will save the reader having to cross reference between section 3.3 and earlier tables and graphs.

Reply: We understand the request for a new figure, but instead adapted Figure 2 to display years as well as distance from top to enable readers to easily connect the plot and sections of the text. We decided to add the distinct U-Th dates (in green) rather than a second axis because of the limited number of samples and their discontinuous nature.

Comment: Line 291 – I'd like to see the further work section expanded with some more detailed suggestions. Making connections between suggested future work and specific issues raised by these results would make this section stronger. It would also be good to have this as a single future work section – at present it is split between the end of the discussion and the end of the conclusion.

Reply: Thank you for a very good idea of splitting the last chapter into two, Future Work and Conclusions This structure provides space for elaborating on detailed suggestions and makes our work more concise.